# Effective Multivalent Oriented Presentation of Meningococcal NadA Antigen Trimers by Self-Assembling Ferritin Nanoparticles

**DOI:** 10.3390/ijms24076183

**Published:** 2023-03-24

**Authors:** Daniele Veggi, Lucia Dello Iacono, Enrico Malito, Giulietta Maruggi, Fabiola Giusti, Panchali Goswami, Werner Pansegrau, Sara Marchi, Sara Tomei, Enrico Luzzi, Matthew James Bottomley, Federico Fontani, Ilaria Ferlenghi, Maria Scarselli

**Affiliations:** 1GSK, Via Fiorentina 1, 53100 Siena, Italy; 2GSK, 14200 Shady Grove Rd, Rockville, MD 20850, USA; 3GSK, Gunnels Wood Rd, Stevenage SG1 2NY, UK

**Keywords:** nanoparticles, ferritin, bacterial antigen

## Abstract

The presentation of viral antigens on nanoparticles in multivalent arrays has emerged as a valuable technology for vaccines. On the nanoparticle surface, highly ordered, repetitive arrays of antigens can mimic their geometric arrangement on virion surfaces and elicit stronger humoral responses than soluble viral antigens. More recently, bacterial antigens have been presented on self-assembling protein nanoparticles and have elicited protective antibody and effective T-helper responses, further supporting the nanoparticle platform as a universal approach for stimulating potent immunogenicity. Here, we present the rational design, structural analysis, and immunogenicity of self-assembling ferritin nanoparticles displaying eight copies of the *Neisseria meningitidis* trimeric adhesin NadA. We engineered constructs consisting of two different NadA fragments, head only and head with stalk, that we fused to ferritin and expressed in *Escherichia coli*. Both fusion constructs self-assembled into the expected nanoparticles as determined by Cryo electron microscopy. In mice, the two nanoparticles elicited comparable NadA antibody levels that were 10- to 100-fold higher than those elicited by the corresponding NadA trimer subunits. Further, the NadAferritin nanoparticles potently induced complement-mediated serum bactericidal activity. These findings confirm the value of self-assembling nanoparticles for optimizing the immunogenicity of bacterial antigens and support the broad applicability of the approach to vaccine programs, especially for the presentation of trimeric antigens.

## 1. Introduction

Self-assembling nanoparticle technologies are well-established and widely used in vaccine research and development. Nanoparticles provide a way to present antigens in repetitive, geometric, and ordered patterns that facilitate B cell activation by cross-linking B cell receptors (BCRs), resulting in an improved immune response [1,2].

The diameter of nanoparticles and the spacing of antigens have been reported to be the most critical parameters for B cell activation, with the optimal diameter of nanoparticles of 10–150 nm and optimal spacing of epitopes (20–25 in number) by 50–100 Å [3,4,5]. Additionally, the multivalent display of antigens on nanoparticles bestows high avidity, crucial for inducing potent immune responses. Further, nanoparticles can present antigens in native-like conformation to the immune system, including specific stabilized conformations since many viral neutralizing antibodies target epitopes available on only one or a few conformations [6].

Several self-assembling nanoparticles displaying viral antigens have been reported. Among these, the 24-meric *Helicobacter pylori* bacterial ferritin protein (12.2 nm in diameter) has been used for the multimerization of HIV-1 envelope (Env) glycoprotein [7,8], Rotavirus VP6 glycoprotein [9], influenza hemagglutinin (HA), and SARS-CoV-2 [10,11,12]. In all cases, these nanoparticles elicited antibodies in immunized animals that were substantially improved in neutralization breadth and potency. In the case of hemagglutinin, the mosaic co-display of diverse hemagglutinins from different influenza virus strains has also been investigated, using insect ferritin that naturally self-co-assembles from heavy and light chains. The mosaic nanoparticle resulted in B-cell responses qualitatively and quantitatively superior to those elicited by homotypic immunogens, due to the promotion of cross-reactive antibody responses [13]. In addition to ferritin, other multimeric self-assembling nanoparticles, e.g., the 60-meric lumazine synthase (LS) from *Aquifex aeolicus* (14.8 nm in diameter) and dihydrolipoyl acetyltransferase (E2p) from *Bacillus stearothermophilus* (23.2 nm in diameter), have been also considered for the design of multivalent HIV-1 immunogens (i.e., gp120 outer domain-LS and gp41-E2p). Further, novel computationally designed nanoparticles [14,15,16] have enabled the multivalent presentation of DS-Cav1, a prefusion-stabilized variant of the RSV F glycoprotein, as well as the receptor binding domain of the SARS-CoV-2 S glycoprotein [17,18].

Recently, protein nanoparticles have also become attractive for the presentation of bacterial antigens. In particular, T4 phage nanoparticles have been used to present *Bacillus anthracis* and *Yersinia pestis* antigens, PA and F1mutV, respectively [19]. Additionally, ferritin fusions have been investigated for enhancing the immunogenicity of gonococcal MtrE [20] and *Borrelia burgdorferi* OspA [21] and a variety of nanoparticles has been used to present the Factor H binding protein of *Neisseria meningitidis* [22]. In all the cases, the nanoparticle-supported immunogens enhanced immunogenicity and protective activity compared to isolated soluble antigens.

Herein, we used ferritin from *H. pylori* to display the trimeric *N. meningitidis* antigen, NadA. Two fragments of NadA were individually fused to ferritin, and each fusion polypeptide self-assembled into nanoparticles, which we analyzed for structure and immunogenicity.

NadA is one of three major recombinant protein components of the 4CMenB vaccine, licensed for use against serogroup B meningococcus (MenB) in all ages from 2 months to 50 years [23]. The *nadA* gene is present in ~30% of pathogenic meningococcal isolates and is often associated with hypervirulent MenB lineages [24].

NadA belongs to the family of trimeric autotransporter adhesins (TAA) that promote adhesion to and invasion of epithelial cells, via the N-terminal region [25,26]. The 3D structures of fragments of NadA (aa 24–170), determined by X-ray crystallography, reveal an N-terminal globular head and an elongated coiled-coil stalk [27,28]. The N-terminal head is the target of most of the bactericidal mAbs isolated from healthy subjects immunized with 4CMenB. Additionally, some of these NadA-head-targeting mAbs inhibited NadA binding to host epithelial cells, suggesting they may inhibit meningococcal colonization [29].

NadA is immunogenic as a soluble recombinant trimeric protein, and for the design of NadA-loaded nanoparticles with enhanced protective immunity, the structural integrity of the antigen is crucial [30]. To further investigate the relationships between the immunogenic properties of NadA and its conformation, we genetically fused the NadA head alone and the head with the stalk, individually, via a flexible linker, to the N-terminus of ferritin. We then evaluated the nanoparticles self-assembled from these fusion polypeptides for the physical stability of the NadA antigens as compared to the soluble trimer. Lastly, we assessed whether the multivalent presentation of NadA on ferritin improves the quality of the immune response as compared to the NadA soluble trimer.

## 2. Results

### 2.1. Design of Ferritin Nanoparticles Constructs Presenting Trimeric NadA

We hypothesized that the ferritin nanoparticle could display NadA in native-like trimeric conformation, with its N-terminal head displayed on the nanoparticle surface, so it can engage B-cell receptors. The N-terminal residues of each ferritin subunits are located proximal to the three-fold symmetry axis, with approximate distances of 28 Å among the alpha Carbon (Ca) atoms [31]. Such disposition makes them optimal insertion sites for trimeric exogenous antigens.

The meningococcal adhesin NadA appeared to be a suitable candidate to test this hypothesis, as it consists of a homotrimer arranged in a coiled-coil stalk that presents a globular head at its N terminus. Two regions of NadA from *N. meningitidis* strain 2996 were considered: (1) the full extracellular domain of the protein (residues 24–342), which includes the stalk and the head, termed NadA_Ext (extended), and (2) the head (residues 24–89), without the stalk, termed NadA_Sht (short). Each of these NadA fragments was genetically fused to the N-terminus of ferritin protein by the amino acid linker SSGAGS, such that the folding and self-assembly of ferritin and the NadA fragments would not be disturbed.

Computer modeling of the self-assembled nanoparticles, with octahedral symmetry, shows eight NadA-extended globular heads (with stalks) spaced 39.4 nm apart and protruding 40 nm from the nanoparticle surface; in contrast, the NadA-short globular heads are more closely spaced at 5.7 nm apart and protrude only 4.7 nm from the ferritin nanoparticle surface (Appendix A).

The NadA_Ext-Fe and NadA_Sht-Fe constructs were expressed in *Escherichia coli* and purified using immobilized metal affinity chromatography and size exclusion chromatography. The size-exclusion chromatography profiles of the constructs showed single peaks, with each eluting earlier than ferritin alone, consistent with them being assembled nanoparticles (Appendix A).

### 2.2. NadA-Ferritin Nanoparticles Form Large and Stable Structures in Solution

To examine the thermostability of the nanoparticles, we collected differential scanning fluorimetry (DSF) profiles. NadA Sht-Fe exhibited a single thermal transition at 87.7 °C, while NadA Ext-Fe showed two melting points at 47.2 and 56 °C (Figure 1 and Appendix A). Ferritin alone had a thermal denaturation that fell outside the instrument’s detectable range (35–95 °C). For this reason, we used differential scanning calorimetry (DSC) to determine the melting temperature (Tm) of ferritin to be 100 °C.

In a previous study, we determined the Tm of the NadA fragment 24–342 (the fragment of NadA_Ext) to be 45 °C [27]. We speculate that the 47.2 °C transition of NadA Ext-Fe could be due to the unfolding of the extended NadA fragment (stalk and head), while the transition at 56 °C could be due to a decrease in the compactness of the ferritin scaffold.

### 2.3. NadA-Ferritin Nanoparticles Are Octahedral Structures

To characterize the structure of NadA_Sht-Fe and NadA_Ext-Fe in more detail, we performed cryo-electron microscopy (Cryo-EM). The NadA_Ext-Fe and NadA_Sht-Fe samples both showed an expected 12-nm-diameter spherical core, consistent with the ferritin scaffold.

Regarding the NadA appendages, the antigen of NadA_Ext-Fe with the extended stalk showed a variety of conformations, possibly because of the flexibility that characterizes long coiled-coil stalks. In contrast, the population of NadA_Sht-Fe was homogeneous and presented shorter protrusions in line with the NadA head being directly connected to ferritin by a hexapeptide linker (Figure 2A).

To determine the high-resolution structure of NadA_Sht-Fe and NadA_Ext-Fe, we analyzed the constructs by Cryo-EM. At 4 Å resolution, NadA_Sht-Fe showed eight well-defined densities located along the three-fold axes of the spherical core ferritin particle (Figure 2B). These densities are consistent with the NadA crystallographic structure [28] since the inner core of the NadA head is composed of three coiled-coil helices surrounded by three elongated wings, tightly packed on the three internal helices. Thus, these data demonstrate that the fusion of NadA 24–89 to ferritin (NadA_Sht-Fe) did not disturb the individual architectures of the two entities composing the nanoparticle, which demonstrates that recombinant-protein self-assembling nanoparticles can present a regular, multi-copy array of this bacterial antigen, preserving its 3D fold and generating the expected multivalent immunogen.

In the case of NadA Ext-Fe, the scaffold core appeared as a well-folded structure formed by 24 ferritin monomers clearly organized in an octahedral arrangement, indicating that the assembly of ferritin was not compromised by the fusion with NadA 24–342. This ferritin scaffold core was surrounded by an additional cloud of electron density with a radius of about 14 nm, originated by the population of highly flexible NadA spikes (Figure 2C,D); however, Cryo-EM data processing could not reconstruct the NadA extracellular domain.

### 2.4. NadA_Ext-Fe Nanoparticles Preserve the Functional Epitopes of NadA

To check whether the functional epitopes on recombinant NadA were conserved on the flexible spikes of NadA_Ext-Fe, we probed these nanoparticles with three recombinant monoclonal antibodies (2C4, 1C6 and 5D11) isolated from peripheral blood mononuclear cells (PBMCs) of healthy adults immunized with 4CMenB vaccine and previously reported as bactericidal and capable of inhibiting the adhesion of meningococci to Chang epithelial cells [29]. These mAbs target the globular head (2D4 and 5D11) or the membrane-distal stalk (1C6) of recombinant NadA, and recognized NadA_Ext-Fe with high affinity and comparable kinetics to recombinant NadA, as shown by surface plasmon resonance (Figure 3), which supports the hypothesis that NadA on the ferritin scaffold maintains the original folding and antigenic repertoire.

### 2.5. NadA-Ferritin Nanoparticles Potently Induce Complement-Mediated Bactericidal Activity and Associated IgG Subclasses

To evaluate how the immune response to NadA epitopes can be affected by presentation on ferritin nanoparticles, we immunized mice with recombinant NadA 24–342, NadA_Ext-Fe, or NadA_Sht-Fe and compared antibody titers (ELISA, Figure 4A) and complement-mediated bactericidal activity (SBA). For the SBA studies, the target bacterium was *N. meningitidis* hypervirulent strain 5/99, a well-established standard for measuring the efficacy of the immune response raised by NadA [32]. Overall, the bactericidal responses indicated that the NadA epitopes were more effective when presented in a multivalent array on the ferritin scaffold. As expected, antibodies raised to recombinant NadA 24–342 (i.e., the extracellular domain) killed MenB strain 5/99 with high efficiency. However, the antibodies raised to NadA_Ext showed a 16-fold increase in bactericidal titer, the highest among all the constructs tested (Figure 4B). This advantage provided by the attachment of the NadA extracellular domain to the ferritin scaffold was also seen with antibodies raised to NadA_Sht-Fe, suggesting that the display of the antigen in a regular array also improved the response to epitopes on the globular head.

To further investigate the reasons for the increased bactericidal efficacy of sera raised to the NadA antigens conjugated to ferritin, we immunized mice with the nanoparticle-conjugated NadA or unconjugated recombinant NadA and analyzed IgG subclasses. Mouse IgG3 and IgG2a are widely recognized to be more effective in inducing complement-mediated bacterial killing compared to IgG1 [33]. Our data show that mice immunized with NadA_Ext-Fe raised the highest IgG2b and IgG3 responses (Figure 5), which could explain the highest bactericidal activity. In particular, the difference between NadA_Ext-Fe and NadA_Sht-Fe, expressed by the geometric mean ratio between the titers of the two groups, was 5.9 fold (95% CI: 2.6–13.6) for IgG2b and 8.6 fold (95% CI: 2.0–35.8) for IgG3.

## 3. Discussion

For more than 200 years, emerging vaccine technologies have successively allowed the development of the next generation of vaccines, including recombinant protein antigens pioneered by the Hepatitis B vaccine [34]. Recombinant subunit vaccines, safe and well tolerated, represent today a well-established alternative to traditional inactivated or live attenuated microorganisms. Moreover, rational antigen design continues to offer new opportunities to enhance the immunogenicity and protective activity of soluble antigens [35].

Besides the addition of adjuvants and the optimization of antigen delivery systems, intense investigation is being dedicated to self-assembling nanoparticles as platforms to display protective antigens. Antigen assembly on nanoparticles results in a regular repetitive disposition of protective epitopes on macromolecular structures whose size promotes efficient uptake by antigen-presenting cells, a critical step to trigger and amplify the adaptive immune response [36,37,38].

In the present study, we investigated the effects on immunogenicity of fusing the meningococcal adhesin, NadA, to self-assembling ferritin from *H. pylori*. NadA was selected in view of its well-known biochemical and immunological properties as a soluble trimeric antigen [28,39], and for the possibility to evaluate the functionality of the antibodies raised by vaccination through the complement-mediated bactericidal assay, a well-established correlate of protection against meningococcal disease [40].

The full-length ectodomain of NadA3 (aa 24–342) is predicted to be highly extended (300–350 Å long) and somewhat flexible. In principle, conjugation to a multivalent carrier could influence the immune response to NadA through multiple mechanisms. The increased antigen size could more efficiently promote the uptake and processing by immune cells. Moreover, the multi-copy, ordered display could help focus the antibody response to the receptor binding region of NadA, located at the N-terminal head. We decided, therefore, to genetically engineer the extracellular domain of this antigen onto the spherical ferritin cage, consisting of twenty-four monomers arranged with octahedral symmetry. Ferritin represented in principle a good carrier to present native NadA spikes, as previous studies demonstrated that this molecule is able to display trimeric viral antigens and elicit significantly higher neutralizing antibody titers compared to soluble trimers [7,8,11].

One of the main issues with displaying antigens on recombinant nanoparticles is the possibility of disassembly following the genetic fusion with complex protein antigens [41]. To explore the impact of antigen size on nanoparticle stability and functionality, we designed two different nanoparticles, one (NadA_Ext-Fe) displaying the full-length extracellular domain of the meningococcal adhesin NadA, consisting of 318 amino acids, and the other (NadA_Sht-Fe) presenting only the N-terminal globular head formed by 65 residues. In the first case, the ferritin scaffold was intended to display the long and flexible appendages beginning with the elongated coiled-coil and ending with the globular head, while NadA_Sht-Fe was expected to present only the relatively compact head domain on the ferritin cage with well-defined orientation and reduced conformational flexibility.

The two constructs consisting of the different NadA fragments fused to the ferritin cage yielded nanoparticles with different stability and homogeneity. Indeed, while the introduction of the short NadA head to the ferritin did not significantly compromise the stability of the spherical cage, which still showed a high Tm (>80 °C), the fusion of the long NadA 24–342 construct resulted in a less thermally stable ferritin nanoparticle, with two distinct transitions at 47.2 and 56 °C. Considering that, in a previous study we reported that the Tm of the recombinant NadA 24–342 fragment was 45 °C [27], which is close to the first Tm of NadA_Ext-Fe. Therefore, we speculate that the bimodal transition of the NadA_Ext-Fe construct may originate from two distinct events, i.e., the unfolding of NadA appendages (Tm = 47.2 °C) and the relaxing of the compact ferritin scaffold (Tm = 56 °C).

The two decorated ferritin nanoparticles also differed in their homogeneity. By EM, the NadA_Sht-Fe construct was homogeneous in terms of particle distribution and shape. However, the NadA_Ext-Fe was much more heterogeneous, with a wide range of conformations, likely as a consequence of the marked flexibility of the full-length NadA. Nonetheless, this nanoparticle was still able to bind bactericidal monoclonal antibodies isolated from human subjects vaccinated with recombinant NadA [29] that target epitopes distributed along the entire NadA structure, suggesting that the structure of protective native epitopes was preserved. Additionally, serological data indicated that the conformational flexibility of NadA_Ext-Fe had no major effect on immunogenicity, as both nanoparticles and the corresponding recombinant NadA fragments elicited comparable titers of antibodies (Figure 4A).

Nevertheless, EM analysis clearly revealed the persistence of globular ferritin in the NadA_Ext-Fe sample population, accompanied by marked flexibility of the extracellular portion of NadA. The wide conformational space experienced by full-length NadA anchored to the ferritin nanoparticle is likely responsible for the smaller size observable in cryo-EM compared to the theoretical diameter of the model in silico where the NadA protrusions are fixed in an elongated conformation which implies the maximum of their extension.

In comparing the in vivo immune responses to our panel of NadA constructs, we observed that antigen multimerization was important not only for the antigen-specific antibody elicitation but even more for the key role in improving the bactericidal antibody level. The increased protective response conferred by the nanoparticle display was clearly appreciable by comparing the bactericidal activity of sera raised by both NadA-ferritin constructs to that induced by recombinant NadA domains alone (Figure 4B). Serum bactericidal activity elicited by NadA_Ext-Fe showed a 16-fold increase compared to NadA_Ext while a 32-fold increase was observable when NadA_Sht-Fe serum was compared to NadA_Sht.

The pronounced structural stabilization derived from the fusion to ferritin could improve the quality of the immune response. We can speculate that the anchoring to the spherical nanoparticle could more closely mimic the native format of NadA, which naturally attaches to the bacterial surface through its carboxyl-terminal transmembrane domain. In such a situation, conformational epitopes could be more stabilized to express their protective potential. A previous study of the head NadA domain, devoid of the coiled stalk, found that it was unable to fold correctly [27]. Our data demonstrate that the globular head is well stabilized by conjugation to ferritin. The gain in conformational stability could, therefore, explain the increased capability to elicit functional antibodies by the NadA_Sht domain.

The remarkable immunogenic properties observed for NadA_Sht-Fe and NadA_Ext-Fe underline the importance of a structure-guided design of NPs to identify the structural requirements that need to be preserved, particularly in the case of antigens with complex three-dimensional folds. Moreover, our study confirms the effectiveness of ferritin nanoparticles in enhancing the immune response against bacterial antigens and supports the use of this strategy for bacterial vaccine targets.

Overall, our data indicate that the fusion of NadA onto the ferritin scaffold results in a clear enhancement of the bactericidal activity conferred by NadA-specific antibodies and provides further evidence in supporting the use of nanoparticles for an efficient presentation of protective bacterial epitopes.

## 4. Materials and Methods

### 4.1. Structure-Based Design

The initial model of a NadA-ferritin fusion was generated in silico by manually placing the C-terminal region of a homology model of full-length NadA3 trimer built starting from crystal structures of NadA5 and NadA3 fragments (PDBs 4CJD and 6EUN) close to the N-terminal region of a trimer of the iron-loaded *H. pylori* ferritin derived from PDB 3EGM. Coordinates were inspected manually using the molecular graphics in PyMOL (www.pymol.org, accessed on 11 February 2019) in order to align the molecules with respect to each other and to evaluate potential linkers for fusion. Linkers were then manually designed as short polypeptide sequences (4–6 aa in length) to span the distance between the input coordinates of scaffold (ferritin) and antigen (NadA). Initially, the NadA3 C-terminal residue (G342) was placed at a 20Å distance from residue D5 of ferritin, while at the same time visually aligning the three-fold axes of the input trimers. Considering measured distances of ~10Å between the three copies of G342 in a NadA3 trimer, and ~28 Å between three D5 residues in a ferritin trimer, a 6 aa-long linker was visually deemed compatible with forming an interface without clashes or significant strains, and especially able to accommodate the predicted angle necessary to fuse the triangle joining NadA3 terminal residues G342 (10Å side) with the triangle merging D5 residues on a ferritin trimer (~28Å side). The SGGAGS sequence was chosen to connect the C-terminal residue of NadA to the N-terminus of ferritin due to its amino acid composition. The central GGAG core was intended to ensure the necessary flexibility to bring in proximity the two molecule ends and two flanking serines were added to increase the hydrophilicity. We concluded, therefore, that the SGGAGS peptide was appropriate to fuse NadA to the ferritin N-terminus. The hexapeptide SGGAGS was generated in silico and merged into a new set of 3D coordinates that included one NadA3 monomer, and one ferritin monomer; symmetry-related copies of this new chimera were generated to recapitulate a trimeric assembly. This new set of coordinates was subsequently energy and geometry minimized in Phenix [42]. Visual inspection of the refined coordinates and analyses of the geometry and slight rearrangements of the linkers provided reassurance that no clashes would likely form between NadA3 and ferritin and that the chosen length would be compatible with the angle required to keep NadA exposed and oriented properly on a ferritin scaffold (Figure 6). Finally, the NadA3 sequence was truncated to generate the same models as above but displaying shorter antigens, namely NadA3 residues 24–342 or NadA_Ext (Extended) and NadA3 24–89 or NadA_Sht (Shortened).

### 4.2. Cloning and Expression

Gene fragments of *nadA* (residues 24–342 and 24–89) were PCR amplified from the *N. meningitidis* serogroup B strains 2996 (NadA3), and were inserted into the pET-15b expression vector (Novagen, Darmastadt, Germany) to enable IPTG-inducible production, as described previously [27]. The sequence numberings used herein refer to the full-length NadA3, with UNIPROT code Q8KH85. The NadA expression constructs were cloned without their signal peptides and with a hexahistidine (6-His) tag and a TEV cleavage site added at the N terminus of each construct to facilitate purification. The gene encoding *H. pylori* strain G27 non-heme iron-containing ferritin (GenBank NP_223316) was synthesized by PCR. The NadA–ferritin fusion genes were generated by fusing *nadA* gene fragments described above to *H. pylori* ferritin (residues 5–167) with an SGGAGS linker, using the polymerase incomplete primer extension (PIPE) cloning methods [43]. After sequencing, each plasmid was transformed into *E. coli* BL21 (DE3) cells (Novagen) for protein production. Expression of NadA and NadA-ferritin constructs was performed in high-throughput complex medium 3 [30 g/L yeast extract, 15 g/L glycerol, 20 g/L K_2_HPO_4_, 5 g/L KH_2_PO_4_, 0.5 g/L MgSO_4_ (pH 7.2)] at 27 °C for 30 h.

### 4.3. Protein Purification

The Nada fusions with ferritin—i.e., the NadA ectodomain with the head and the stalk, NadA_Ext-Fe (residues 24–342) and the NadA head, NadA_Sht-Fe (residues 24–389)—were expressed and purified as described in [27]. Both forms with ferritin genetically linked, as well as ferritin alone, were expressed as a His-tag fusion in *E. coli* BL21 (DE3) cells (New England Biolabs, Ipswich, MA, USA). Cell pellets were resuspended in binding buffer (300 mM sodium chloride, 50 mM sodium phosphate, pH 8) and lysed by sonication (Qsonica Q700, Newtown, CT, USA) with five cycles of 30 s of sonication (40% amplitude) interspersed with 1 min on ice. Cell lysates were clarified by centrifugation at 36,200× *g* for 45 min and then affinity chromatography was performed at room temperature using a HisTrap HP 5 mL linked to an AKTAPurifier (GE Healthcare, Chicago, IL, USA), with the protein being eluted by employing an imidazole gradient. Then, to remove the His-tag, the samples were incubated overnight with TEV in a ratio of 5:1. Subsequently, size-exclusion chromatography (SEC) was performed using a Superose 6 10/300 GL column (GE Healthcare), equilibrated with running buffer (PBS). Bands at the expected molecular weight of NadA_Ext-Fe, NadA_Sht-Fe and ferritin alone were observed in SDS-PAGE analysis, using NuPAGE Novex Bis-Tris 4–12% gels ran in MES buffer, then stained with SimplyBlue SafeStain (ThermoFisher, Whaltam, MA, USA). Fractions were then pooled and filtered using a Millex 0.22 μM filter.

### 4.4. Cryo-Electron Microscopy (Cryo-EM) Sample Preparation

Glow-discharged Quantifoil R2/2 grids (Electron Microscopy Sciences, Hatfield, PA, USA) were charged with 2.3 μL of NadA_Ext-Fe or NadA_Sht at 0.2 mg/mL in a Vitrobot blotting robot (ThermoFisher Scientific) using 5 s blotting time with 100% humidity, and then plunge-frozen in liquid ethane cooled by liquid nitrogen.

### 4.5. Cryo-EM Data Acquisition

Automated data collection was performed at the Nanoscience laboratories in Cambridge as part of the Krios Consortium. Movie stacks were collected with EPU on a Titan Krios (Thermo Fisher Scientific) operated at 300 kV and equipped with a BioQuantum energy filter operated with a 20 eV energy slit with a Falcon III direct detection camera (FEI) using EPU software. Images were recorded at a nominal magnification of 75,000×, corresponding to a pixel size of 1.072 Å. Each image stack contains 31 frames recorded in counting mode every 0.20 s giving an accumulated dose of 30 electrons/Å^2^ and a total exposure time of 60 s. Images were recorded with a set defocus range of −1.8 to 0.5 μm.

### 4.6. Cryo-EM Data Processing

MotionCor2 (Zheng et al. 2017) was run to align and dose-weight the collected movie micrographs. Automatic particle picking using RELION-3.0 (RRID: SCR_015701) [44] allowed the extraction of particles in 200 × 200 pixel boxes, later classified into 2D classes. A subset of 2D class averages was selected and used to generate reference models with the application of octahedral symmetry. Several rounds of 3D classification and refinement were used to sort out a subpopulation of particles that went into the final 3D reconstructions. Octahedral symmetry restraints were applied for all 3D refinement and classification steps. A soft solvent mask around the nanoparticle core was introduced during the final 3D classification, refinement, and post-processing. The resolution of the final reconstructed map was 4.2 Å (FSC = 0.143). The resulting EM maps have been deposited to EMDB with IDs D_1292127947. Single Particle Processing workflow for the NadA_Sht-Fe dataset is shown in Appendix A. The program Chimera (https://www.cgl.ucsf.edu/chimera/, 17 December 2021) was then employed to perform rigid body fitting using the crystallographic coordinates of the ferritin alone (PDB 3EGM) into the electron density maps.

### 4.7. Surface Plasmon Resonance (SPR)

SPR experiments were performed using a Biacore T200 instrument (GE Healthcare). mAbs (50 µg/mL in 0.01 M HEPES pH 7.4, 0.15 M NaCl, 3 mM EDTA, 0.005% *v*/*v* Surfactant P20 (HBS-EP+, GE Healthcare) were captured on the second flow cell of a CM5 sensor chip (GE Healthcare) coated with polyclonal anti-human Fc antibodies until a capture level of ca. 3000–3500 RU was reached. Analytes were injected for 60 s and with 60 s intervals at 30 µL/min, applying antigen concentrations of 6.25, 12.5, 25, 50 and 100 nM in HBS-EP+. Finally, the dissociation of mAb-antigen complexes was monitored for an additional 300 s. In most cases, mAb-antigen complexes were too stable to permit quantitative kinetic analysis. However, where possible, kinetic analysis was performed using the Biacore Evaluation Software version 3.0, applying the 1:1 binding model of interaction.

### 4.8. Differential Scanning Fluorimetry (DSF)

Real-time simultaneous monitoring of fluorescence at 330 nm and 350 nm during the thermal ramp of native nanoparticles was carried out in a Prometheus NT.48 instrument from NanoTemper Technologies with an excitation wavelength of 280 nm. Capillaries were filled with 10 μL of a suspension of nanoparticles (0.5 mg/mL in PBS), placed into the sample holder and the temperature was increased from 25 to 95 °C at a ramp rate of 1 °C/min, with one fluorescence measurement per 0.044 °C. The ratio Em_350nm_/Em_330nm_, which takes into account the change in tryptophan fluorescence intensity and the shift in maximum emission to higher or lower wavelengths was plotted as a function of the temperatures. The fluorescence intensity ratio and its first derivative were calculated with the manufacturer’s software (PR.ThermControl, version 2.1.2). Three independent measurements were carried out for each sample.

### 4.9. Differential Scanning Calorimetry (DSC)

The thermal stability of ferritin nanoparticles was assessed by differential scanning calorimetry (DSC) using a MicroCal VP-Capillary DSC instrument (GE Healthcare). Ferritin samples were prepared at a protein concentration of 0.4 mg/mL in PBS buffer. The DSC temperature scan ranged from 20 °C to 110 °C, with a thermal ramping rate of 200 °C per hour and a 4 s filter period. Data were analyzed by subtraction of the reference data for a sample containing buffer only, using the Origin 7 software. All experiments were performed in duplicate, and mean values of the melting temperature (T_m_) were determined.

### 4.10. Mice Immunizations

To prepare antisera, 20 μg of purified recombinant proteins, NadA_Ext and NadA_Sht, assembled or not into ferritin nanoparticles, were used to immunize 6–8-week-old CD1 female mice (Charles River Laboratories, Wilmington, NC, USA); 5–7 mice per group were used. The antigens together with aluminum hydroxide (3 mg/mL) were administered intraperitoneally on days 1, 21, and 35. Blood samples for serological analysis were taken on day 49. The treatments were performed in accordance with internal animal ethical committee and institutional guidelines.

### 4.11. Complement-Mediated Bactericidal Activity

Serum bactericidal activity against *N. meningitidis* strain 5/99 was evaluated as described previously [45,46], with pooled human serum obtained from volunteer donors under informed consent used as a complement source. Briefly, *N. meningitidis* strain 5/99 was grown overnight on chocolate agar plates at 37 °C in 5% CO_2_. Colonies were inoculated in Mueller–Hinton broth containing 0.25% glucose and incubated at 37 °C with shaking until reaching an optical density of 0.25 at 600 nm. The bacteria were diluted in Dulbecco’s saline phosphate buffer (Sigma or equivalent) with 0.1% glucose (Sigma or equivalent) and 1% BSA (Sigma or equivalent) at the working dilution of 10^4^ CFU/mL. All mouse sera to be tested were heat inactivated for 30 min at 56 °C. The total volume in each well was 50 μL, with 25 μL of serial twofold dilutions of test serum, 12.5 μL of bacteria at the working dilution, and 12.5 μL of human plasma. The plate was incubated for 1 h at 37 °C; 7 μL of each sample was spotted in duplicate on Mueller–Hinton agar plates. The agar plates were incubated for 18 h at 37 °C, and the colonies in each spot were manually counted. Controls included bacteria incubated with complement and immune sera incubated with bacteria and with complement inactivated by heating at 56 °C for 30 min. Bactericidal titers are defined as the reciprocal serum dilution that gives a 50% decrease in CFU after 60 min incubation in the reaction mixture, compared with the mean number of CFU in the control reactions at time 60. The bactericidal titers reported in this study are related to pooled mouse sera.

### 4.12. ELISA

Microtiter plates were coated overnight at 4 °C with 0.015 μM purified recombinant NadA (full-length ectodomain) [39]. Plates were incubated with mice sera followed by alkaline phosphatase-conjugated anti-mouse antibodies. After the addition of *p*-nitrophenyl phosphate, optical density was analyzed using a plate reader at a dual wavelength of 405/620–650 nm. Total IgG was quantified in ELISA units/mL by comparison to the internal standard curve. IgG subclasses were quantified as the dilutions of sera that gave an optical density at 450 nm (OD_450_) of 0.4.

### 4.13. Statistical Analysis of ELISA Data

A variance analysis model (ANOVA), including the group as a categorical fixed factor and allowing for heterogeneous variances through the groups, was applied to the Log10 titer by time point.

The estimated arithmetic means of Log10 titer were obtained for each group from the variance analysis model applied to each time point. The arithmetic mean for Log10 titer ($\hat{\mu }$) was back-transformed into a geometric mean (GMT) for titer, by using the following formula:(1)$$\hat{\mathrm{GMT}}=10^{\hat{\mu }}.$$

Graphs, presenting the individual values as well as the corresponding geometric means and 95% confidence intervals, were built for each immunization group.

Each difference in arithmetic mean for Log10 titer between chosen groups (${\hat{\mu }}_{1-}{\hat{\mu }}_{2}$, where ${\hat{\mu }}_{1}$ and ${\hat{\mu }}_{2}$ represent the estimated arithmetic means of the first and second groups, considered in the comparison, respectively) was estimated from the same variance analysis model. The corresponding two-sided 95% confidence limits were computed for these differences.

The difference in arithmetic mean was back-transformed into a ratio of geometric means (GMR) for titer, by using the following formula:(2)$$\hat{\mathrm{GMR}}=10^{{\hat{\mu }}_{1-}{\hat{\mu }}_{2}}.$$

The 95% confidence limits for GMR of titer were calculated by applying a similar back-transformation to the above-mentioned lower and upper confidence limits for the difference in arithmetic mean for Log10 titer:(3)$$95\%\;\mathrm{Lower}\;\mathrm{Confidence}\;\mathrm{Limit}\;\mathrm{for}\;\hat{\mathrm{GMR}}=10^{95{\%\;\mathrm{Lower}\;\mathrm{Confidence}\;\mathrm{Limit}\;\mathrm{for}\;\hat{\mu }}_{1-}{\hat{\mu }}_{2}},$$
(4)$$95\%\;\mathrm{Upper}\;\mathrm{Confidence}\;\mathrm{Limit}\;\mathrm{for}\;\hat{\mathrm{GMR}}=10^{95{\%\;\mathrm{Upper}\;\mathrm{Confidence}\;\mathrm{Limit}\;\mathrm{for}\;\hat{\mu }}_{1-}{\hat{\mu }}_{2}}.$$

## Figures and Tables

**Figure 1 ijms-24-06183-f001:**
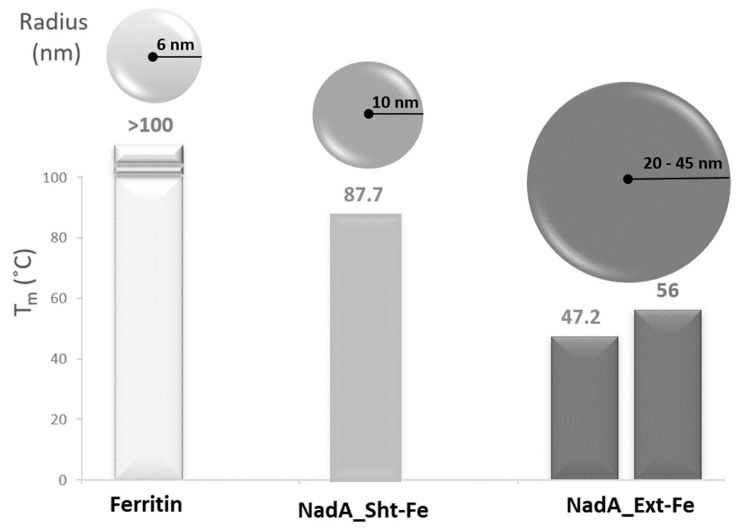
Size and thermostability of NadA-ferritin constructs. The sizes (radii) of the NadA-ferritin nanoparticles were estimated by Cryo-EM as described in the text. The NadA_Ext-Fe radius is shown as a range estimated from EM and the maximum theoretical in silico–predicted value (Appendix A). Melting temperatures (Tm °C) estimated by DSF are reported on top of the corresponding histograms.

**Figure 2 ijms-24-06183-f002:**
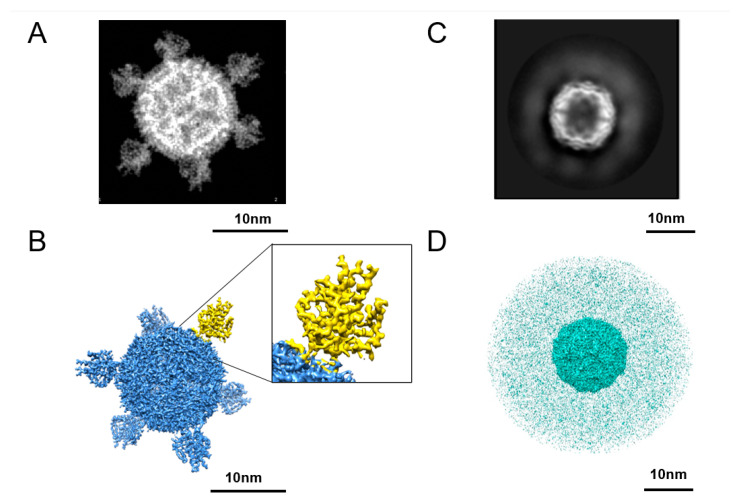
Cryo-EM characterization of head-only NadA_ferritin (**A**,**B**) and full-length NadA_ferritin (**C**,**D**). (**A**) 2D class average showing the icosahedral ferritin scaffold decorated by eight NadA heads (only 6 of them are visible). (**B**) Reconstructed 3D map of the NadA-ferritin head structure at 4 Å of resolution with ferritin scaffold and heads (in blue) and one of the heads in yellow. Close-up view of one NadA head showing the folding of residues composing the NadA head. (**C**) 2D class average of NadA-ferritin FL showing the icosahedral ferritin scaffold correctly folded, decorated by blobs of densities corresponding to the highly flexible and elongated stalk of the NadA homotrimers. (**D**) Reconstructed 3D map of full-length NadA_ferritin (in cyan) showing the inner ferritin scaffold surrounded by a cloud of electron density generated by the flexibility of the NadA stalks.

**Figure 3 ijms-24-06183-f003:**
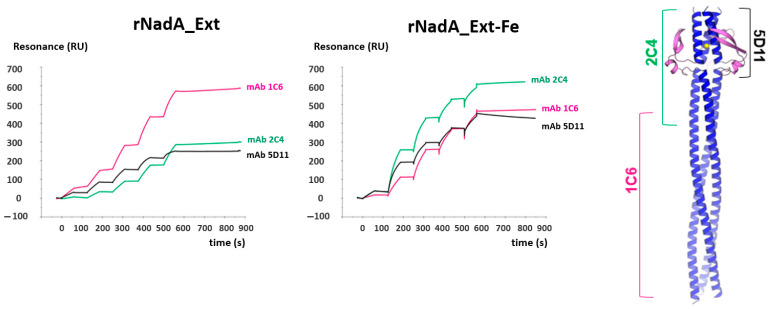
mAb recognition of conformational epitopes on NadA ectodomain, uncoupled and coupled to ferritin scaffold. (**Left**) SPR sensorgrams showing the binding of mAbs that recognize conformational epitopes on the N-terminal globular head domain (2C4, green SPR profile; 5D11, black profile) and stalk (1C6, pink profile) of free recombinant NadA (rNadA) and NadA-ferritin (NadA_Ext-Fe) RU = resonance units; s = seconds. (**Right**)—Mapping of antibodies onto the NadA structure as reported by Giuliani et al. [29].

**Figure 4 ijms-24-06183-f004:**
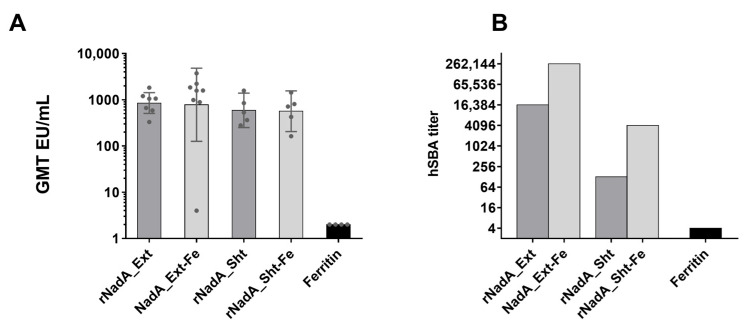
Immune responses elicited by NadA-ferritin nanoparticles. (**A**) Geometric mean titers of IgG induced by different constructs as detected by ELISA. (**B**) Increased serum bactericidal activity in the presence of human complement (hSBA) elicited by both nanoparticles (NadA_Ext-Fe and NadA_Sht-Fe) compared to soluble NadA. hSBA is expressed as the reciprocal of highest dilution of pooled-mice immune sera able to induce killing of 50% of meningococci in vitro.

**Figure 5 ijms-24-06183-f005:**
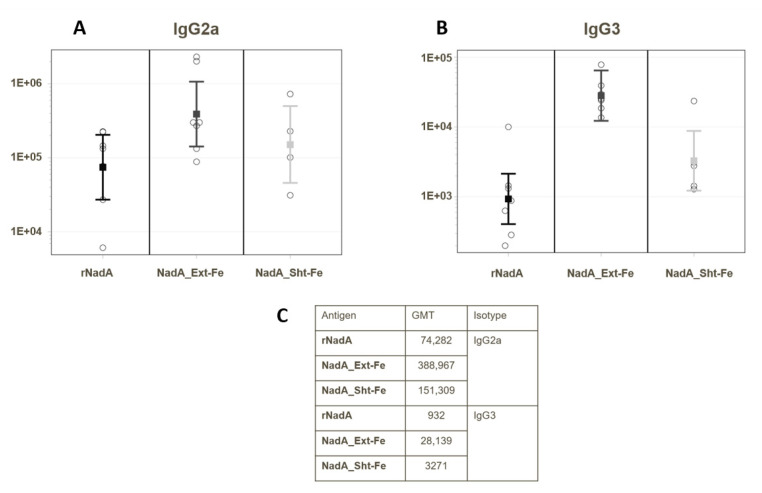
Immunoglobulin subclasses induced by immunization with rNada, NadA_Ext-Fe, and NadA_Sht-Fe. Geometric mean titers (GMT) of (**A**) IgG2a and (**B**) IgG3 induced in mice by vaccination with uncoupled recombinant NadA (rNadA) and the two ferritin-coupled NadA constructs. (**C**) Values of the GMT data plotted in panels (**A**,**B**). Mouse ID 75 was excluded from the analysis as an outlier (Dixon’s test for a single outlier was applied).

**Figure 6 ijms-24-06183-f006:**
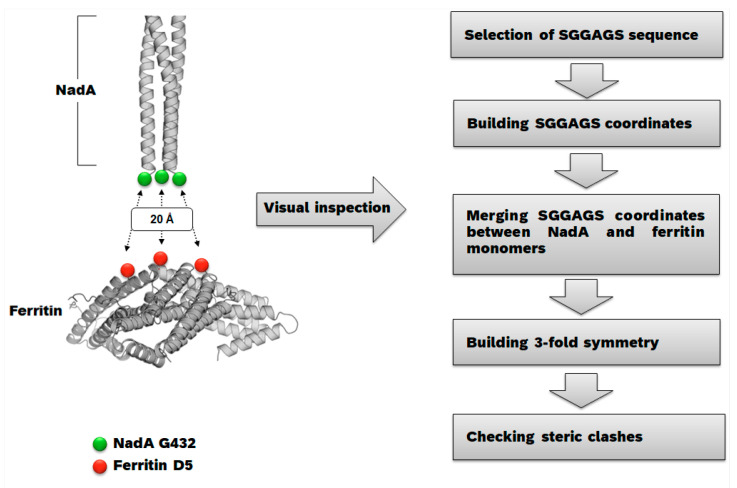
Flow diagram recapitulating the nanoparticle design procedure.

## Data Availability

The data that support the findings of this study are available in the methods and Appendix A of this article.

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
