# Peer review of "Effective Multivalent Oriented Presentation of Meningococcal NadA Antigen Trimers by Self-Assembling Ferritin Nanoparticles"

_ijms, 2023, doi:10.3390/ijms24076183_

Round 1
Reviewer 1 Report
Review of "Effective multivalent oriented presentation of meningococcal NadA antigen trimers by self-assembling Ferritin nanoparticles," by Veggi, et al. as an Article for the International Journal of Molecular Sciences.
The authors report an investigation of a system for displaying viral antigens as use as a vaccine. They use a virion surface, ferritin, which possesses a regular structure with controlled spacing, which is critical for antigen presentation for a good immune response.
Ferritin self assembles into a capsid and thus can be used as a system to self-assemble an antigen presenting surface. They use ferritin to present the NadA antigen for meningococcus B using expression. Their system can display 8 copies of NadA on the ferritin. They use cryo-EM to characterize the structure, which are supported by computer modeling results. They characterize the thermostability of the particles using differential scanning fluorimetry (DSF), differential scanning calorimetry (DSC), and probe the functional epitopes. They measure the bactericidal activity and the corresponding immune response in vivo in mice.
The authors good description of the context of their work which gives the reader a good understanding of the novelty of the work. The data supports their conclusions, and results of the work could positively impact vaccine development.
There are only minor issues that need to be addressed before publication:
1.) Figure 3 seems to be cut off, where part of it is not visible.
2.) Error bars should be included on the data in Figure 4.
3.) The plot needs to have the axes labeled in Figure 3 with the relevant quantity (i.e., time (s)).
4.) The amino acid linker on line 112 should be written in caps (SSGAGS).
Author Response
Dear Editor,
please find below the point-by-point reply to reviewer #1:
- Figure 3 seems to be cut off, where part of it is not visible.
- AU tresponse: Figure 3 has been resized as requested
2.) Error bars should be included on the data in Figure 4.
- AU response: Errors bars have been added in panel A of figure 4, where GMT of IgG titers are reported. Panel B shows instead single shot hSBA titers from pooled sers, so it was not possible in this case include error bars.
3.) The plot needs to have the axes labeled in Figure 3 with the relevant quantity (i.e., time (s)).
- AU response: Axes of Figure 3 have been labelled as requested
4.) The amino acid linker on line 112 should be written in caps (SSGAGS).
- AU response: The linker has been written in caps as requested

Reviewer 2 Report
The authors of this manuscript investigated the immunogenicity of a multivalent antigen designed with fused multimers of NadA, a meningococcal adhesin, and H. pylori derived ferritin. An in vitro study data presented shows that their multivalent antigen construct exhibited higher bactericidal activity mediated by NadA-specific antibodies compared to the native NadA antigen alone. The manuscript is well written, and the results and discussion are well described. However, I have the following suggestions:
Major:
1. The methodology section is complicated to understand, especially the structure-based design part. The authors could have introduced a schematic diagram to explain their design process.
2. The authors need to clarify that they obtained cryoEM data using SerialEM, EPU, or both.
3. The SPR sensogram output lacks a dissociation phase and values of Kon and Koff which are crucial to understand binding kinetics.
Minor:
1. The authors should explain why they selected the SGGAGS linker.
2. The defocus range of 1.2 to 3.4 uM seems too high to resolve atomic information.
3. The authors should specify the source of the mAb used in SPR.
4. Some statements are complicated and need to be simplified (e.g., line 475) and the use of half brackets may confuse readers (i.e. line 109-111)
Author Response
Dear Editor,
please find below the point-by point reply to reviewer #2:
Major:
- The methodology section is complicated to understand, especially the structure-based design part. The authors could have introduced a schematic diagram to explain their design process.
- AU response: A schematic diagram has been introduced in the methodology section as requested
- The authors need to clarify that they obtained cryoEM data using SerialEM, EPU, or both.
- AU response: We clarified that CryoEM data acquisition has been carried out using only EPU (lines 409-412)
- The SPR sensogram output lacks a dissociation phase and values of Konand Koff which are crucial to understand binding kinetics
- AU response: The dissociation phase is not absent but intentionally left short (550-850s) as the complexes formed, with perhaps one exception, are far too stable for meaningful kinetic analysis. With the exception of the couple 5D11/rNadA_Ext-Fe, over the course of the initial 300 sec no measurable dissociation can be observed. Hence, the sensograms are shown to illustrate the extraordinary stability of the interaction without further kinetic analysis
Minor:
- The authors should explain why they selected the SGGAGS linker.
- AU response: To explain why the SGGAGS peptide was selected, we added the following sentences in the Materials and Methods section: “The SGGAGS sequence was chosen to connect the C-terminal residue of NadA to the N-terminus of ferritin due to its amino acid composition. The central GGAG core was intended to ensure the necessary fexibility to bring in proximity the two molecule ends and two flanking serines were added to increase the hydrophilicity. We concluded therefore that the SGGAGS peptide was appropriate to fuse NadA to the ferritin N-terminus.” (lines 353-358)
- The defocus range of 1.2 to 3.4 uM seems too high to resolve atomic information.
- AU response: There was a mistake in the value of defocus and we thank the reviewer for having highlighted it. The correct defocus range (-1.8 to 0.5 um) has been now reported (line 410)
- The authors should specify the source of the mAb used in SPR.
- We specified the source of mabs used in the SPR measurements as following: “ we probed these nanoparticles with three recombinant monoclonal antibodies (2C4, 1C6 and 5D11) isolated from peripheral blood mononuclear cells (PBMCs) of healthy adults immunized with 4CmenB vaccine and previously reported as bactericidal and capable of inhibiting the adhesion of meningococci to Chang epithelial cells [29].” (lines 182-185)
- Some statements are complicated and need to be simplified (e.g., line 475) and the use of half brackets may confuse readers (i.e. line 109-111)
- The original sentence “The ratio of the recorded emission intensities (Em350nm/Em330nm), which represents the change in tryptophan fluorescence intensity as well as the shift of the emission maxi-mum to higher wavelengths (“red-shift”) or lower wavelengths (“blue-shift”) was plotted as a function of the temperatures” was simplified in “The ratio Em350nm/Em330nm, which takes into account the change in tryptophan fluorescence intensity and the shift of maximum emission to higher or lower wavelengths was plotted as a function of the temperatures.” Moreover, half brackets has been removed through all the manuscript
